# Enhancing the Long-Term Ecological Management and Monitoring of Landscapes: The L-TEAM Framework

**Mystyn Mills** [1,2,*], **Loralee Larios** [2] **and Janet Franklin** [2,3]

1    Department of Geography, California State University Long Beach, 1250 Bellflower Blvd.,
     Long Beach, CA 90840, USA
2    Department of Botany and Plant Sciences, University of California, Riverside 900 University Ave.,
     Riverside, CA 92521, USA; loralee.larios@ucr.edu (L.L.); jfranklin2@sdsu.edu (J.F.)
3    Department of Geography, San Diego State University, 5500 Campanile Dr., San Diego, CA 92182, USA
*    Correspondence: mystyn.mills@csulb.edu

**Abstract:** Long-term monitoring and adaptive ecological management are essential to the conservation of biodiversity. Yet, achieving successful long-term ecological monitoring and management, especially at the landscape level, has proven challenging. In this paper, we address the hurdles faced in sustaining long-term monitoring and management for landscape-scale efforts by offering three promising conceptual and methodological developments that support such initiatives. Then, we introduce L-TEAM, a long-term ecological adaptive monitoring and management framework that integrates those three components using four tools: a conceptual model, clearly defined and measurable objectives, scientifically robust experimentation, and decision support tools. Finally, using a case study, we demonstrate L-TEAM's effectiveness in supporting the long-term monitoring and management of a landscape conservation project with diverse habitat types and multiple management objectives. This structured decision framework not only facilitates informed decision making in management practices, but also ensures the implementation of scientifically grounded long-term monitoring. Additionally, L-TEAM holds the potential to enhance our understanding of ecosystem functioning and biodiversity responses to disturbances and management actions.

**Keywords:** long-term ecological monitoring; adaptive monitoring; landscape conservation; ecological management; state-and-transition models

## 1. Introduction

Landscape-scale conservation projects present especially challenging circumstances for ecological management and monitoring [1]. Often, these projects comprise complex and non-equilibrium systems that are made up of multiple habitat types that vary over both space and time and must accommodate multiple uses and management goals. Comprehensive, adaptive, long-term ecological monitoring (spanning longer than 10 years) is crucial to help determine what strategies and methods can effectively conserve and restore these systems [2].

Conservation land managers are confronted by many questions, including but not limited to what, how, where, and when to manage and monitor [3]. Because the management and monitoring of landscapes must accommodate a variety of soil and vegetation complexes, as well as seasonal and annual patterns of precipitation, and variation in disturbance regimes, there are often multiple answers to these questions. These projects often must also balance management for human use and resource procurement with restoration and conservation goals, which requires careful assessment of the impacts of such activities on the success of restoration and conservation measures [4–6]. Thus, conservation management at the landscape level requires both spatially and temporally explicit approaches for implementing and monitoring management practices.

Ultimately, the management of ecological systems should be linked to scientifically informed monitoring [7,8]. This requires clearly defined questions regarding specific management actions and conservation goals that are tractable. These questions and goals should be established early on, and they should help guide what is being monitored and why (i.e., indicators), as well as clearly articulating how success is measured [9]. Both management and monitoring should be adaptive, in that as new information or technologies are acquired or new questions arise, monitoring and management actions can be updated accordingly. Moreover, a system should be in place for communicating understanding of system dynamics to future project members; this requires clear and accessible records of past experiments and findings so that new stakeholders do not "reinvent the wheel" or succumb to past mistakes. Thus, the question is how we can provide restoration and conservation scientists, land managers, and other stakeholders with a systematic approach to collaboratively design and implement a long-term, adaptive ecological monitoring (L-TEAM) program.

The following sections review the challenges to long-term monitoring and management, and introduce some conceptual and methodological developments that hold promise in their ability to address the challenges of defining goals, targets and indicators, and designing good experiments for testing and improving the long-term ecological monitoring and management of heterogeneous landscapes.

## 1.1. Challenges to Adaptive Ecological Management and Monitoring

While long-term monitoring has been acknowledged as a crucial part of conservation actions, several factors make it difficult to implement and maintain a long-term ecological monitoring program. Likens and Lindemayer [10] conducted an extensive review of long-term ecological monitoring and management projects. They present several examples of long-term ecological monitoring programs such as the Hubbard Brook Ecosystem Study and the Alberta Biodiversity Monitoring Program (ABMP) [10]. They identified key factors that contributed to successful monitoring and were often absent from unsuccessful projects (Table 1). These included (1) clear management goals and questions linking monitoring to those goals, (2) detailed conceptual models, (3) sound experimental designs, and (4) relevance to management objectives and targets. Others [11–14] have conducted their own reviews and come to similar conclusions. Likens and Lindenmayer [10] also argue for an adaptive monitoring approach wherein question setting, experimental design, data collection, analysis, and interpretation take place iteratively. With an adaptive approach, findings from well-designed experiments are used to update management approaches and potentially goals, to modify or generate new questions regarding the efficacy of management, and to improve overall understanding of systems and how to manage them effectively.

Many problems faced by long-term and landscape-level projects stem from the lack of clearly articulated questions and goals at the outset [15]. Without driving questions and goals, monitoring may proceed haphazardly. This often includes poorly designed (or a lack of) treatment and control sites, a lack of consideration of the statistical power to detect trends, and inconsistent or poorly communicated management and data collection protocols. Moreover, without clearly defined questions and goals, there is often disagreement over what should be monitored, which can lead to the poor monitoring of many things poorly instead of the effective monitoring of a few things [7]. Alternatively, if an indicator species or other proxy is chosen, there tends to be a lack of a quantified relationship between the entity and the process(es) for which they are surrogates [10,16].

Projects often also face constraints related to funding and time. Research, when it is conducted, is often carried out on timescales related to graduate programs and grant duration (i.e., 3–5 years). Additionally, while funding for monitoring may be included in project and research proposals, by the time monitoring is implemented, funds may be sparse [7]. This often results in an ad hoc strategy with a focus on easy to assess, short-term ecological indicators that may or may not fully relate to the recovery and function of

an ecosystem, let alone answer outstanding questions about the efficacy of management actions [2,10]. A key outcome of the United States' National Science Foundation Long-Term Ecological Research program is that researchers can extend their planning schedule to gain a more nuanced understanding of the system, which could generate more informed future monitoring and research decisions [17].

**Table 1.** Elements of successful long-term monitoring and management projects. Adapted from [10].

| Key Element | Importance | Key References |
| --- | --- | --- |
| Clear Management Goals and Questions | Helps establish quantifiable objectives for measuring progress, can change as new data become available | [18] |
| Detailed Conceptual Model | Guides the development of questions, clearly communicates what is known about a system, identifies areas where knowledge is lacking | [19,20] |
| Sound Experimental Design | Statistically and biologically sound study designs are key to rigorously answering questions and assessing whether objectives have been met and/or the effectiveness of management actions | [11,21] |
| Relevance to Management Objectives and Targets | Clear communication amongst stakeholders and consolidation of knowledge across stakeholders, ensures objectives, study designs, and management actions are appropriate and meet the needs/requirements of all entities involved | [22,23] |

Additionally, successful management also depends on diverse knowledge sources to understand and articulate system dynamics and ecological processes and mechanisms [8,21]. Often, this knowledge is spread across projects, stakeholders, and documents. It can be difficult to communicate among stakeholders and with the public, and importantly, it can be challenging to update management plans and practices when new information or technologies become available. Hence, there is a need for a clear, concise and systematic way to communicate among stakeholders that can both identify where new information is needed and where it can be updated once it is obtained.

While a foundation for adaptive ecological monitoring has been established [10,15] and the benefits are evident [15,24], there has yet to be comprehensive guidance on how to implement an adaptive monitoring program using a standardized approach, especially at the landscape level. Even among the successful projects reviewed by Likens and Lindenmayer [10], there is considerable disparity among approaches. Here, we present a systematic, holistic approach to developing long-term ecological adaptive monitoring and management (L-TEAM). We first discuss three promising conceptual and methodological developments that can support long-term ecological monitoring and management: (1) state-and-transition models, (2) objective-oriented goal development, and (3) decision support tools. Then, we integrate these procedural and analytical tools for the first time into a novel framework for applying principles from long-term ecological monitoring and management to the practice of managing heterogeneous landscapes to meet conservation goals. Finally, we present the application of the L-TEAM framework to a case study of a Southern California riparian shrubland.

### 1.2. Conceptual and Methodological Developments Thus Far

#### 1.2.1. Conceptual Models of Ecological Systems

Lindenmayer and Likens [10] and others consider conceptual models essential to successful adaptive management [8,20,25]. Conceptual models are typically graphical representations of concepts that describe a current understanding of the fundamental principles and processes of a system and the relationships among its parts [20]. Ideally, con-

ceptual models should enhance understanding of the system and facilitate communication amongst stakeholders (for an example from conservation monitoring see [11,26].

State-and-transition models (STMs) are one approach to formal conceptual representations of dynamic ecological systems. Westoby (1989) pioneered the use of STMs when it became apparent that traditional plant successional theories that proceeded linearly toward a single climax community were inadequate to describe many ecosystems recovering from disturbance, especially in semi-arid terrestrial systems [19,27,28]. These models improved upon the more linear model by providing for multiple successional pathways, multiple steady states, and multiple thresholds for transitions between states. The L-TEAM framework uses STMs to conceptually represent the ecological systems under management.

Consistent terminology is critical to facilitate comparison and communication across projects [1]. Lack of consistency in the past has led to criticism of STMs. Stringham at al.'s [1] proposed definitions help clarify what each component of the model should represent and assist with communicating the model results. *States* are recognizable, resistant and resilient complexes of soil and vegetation structure (See Figure 1 large boxes). *Thresholds* are points in space and time where, once crossed, the key ecological processes responsible for a system's identity degrade past a point of self-repair, and active restoration is then required to restore the previous state (See the line bisecting the arrows between the boxes in Figure 1). *Transitions* are pathways between states (See the arrows in Figure 1). Transient and irreversible transitions are often triggered by natural or human-caused disturbances which may occur quickly, as with a fire or flood, or more slowly in response to repeated forms of stress such as grazing or drought. *Alternate states* are the long-term persistence of different plant and soil complexes on an alternative trajectory than the state of interest.

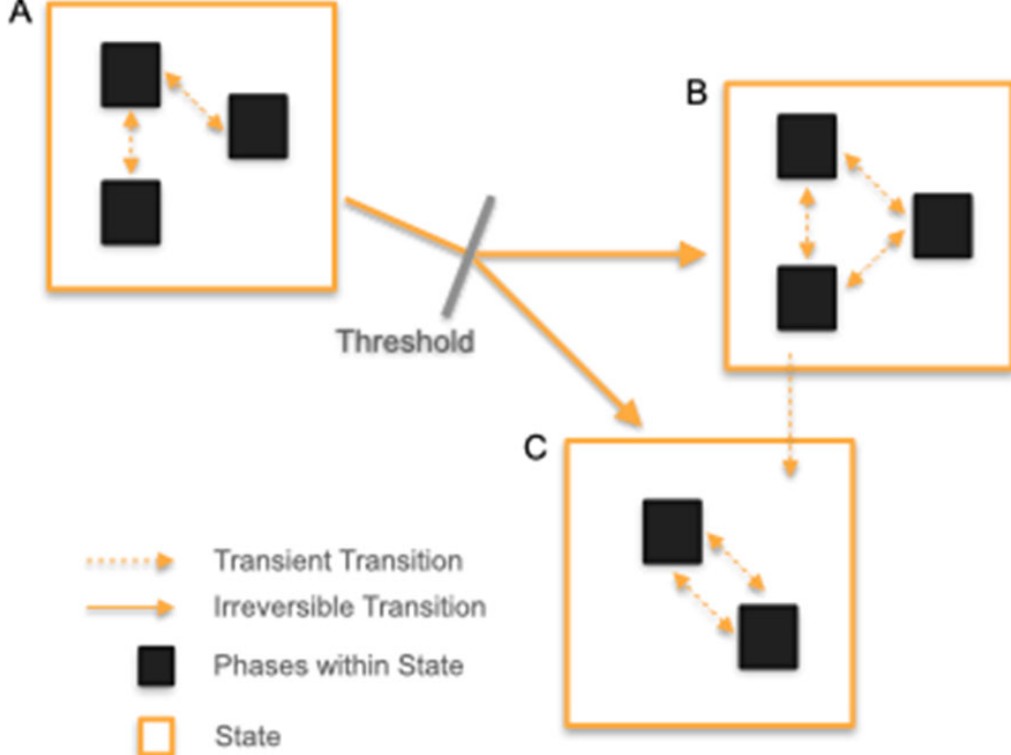

**Figure 1.** A simplified conceptual state-and-transition model. The large boxes represent states. (**A**) is the original state and (**B,C**) are alternative states. The smaller solid boxes represent phases within a state, and the arrows indicate directional transitions between states with known or hypothesized drivers. The line bisecting the arrows between the larger boxes represents a threshold that once crossed, often requires active restoration approaches to return the system to the desired state. Inspired by [1].

Being inherently stable, state changes are only possible when a threshold has been crossed. Within each vegetation state, there is often the potential for large variation in species composition that can be accommodated in the concept of "phase-shifts" which are defined as plant community dynamics within a state [1]. For a true state change, the system must cross a boundary or threshold that results in changes in a site's "identity" (i.e., its underlying processes, moving from A to B in Figure 1), resulting in different sets of potential plant communities, for example, from grasslands to shrublands [1]. With the addition of phase shifts, successional trajectories can be recognized and incorporated into these models (See Figure 1 smaller, solid boxes). Unlike state changes, phase shifts align with a site's natural disturbance regime and (successional) trajectory; they fall within the site's "identity".

In landscape conservation, the difference between phases and states is key, and likely necessitates different management approaches for each. For example, conservation objectives may dictate that all phases in a target state are present, especially if species of conservation concern are associated with specific phases. Moreover, if natural disturbances (for example, a natural flooding regime) are no longer operating to drive shifts between phases, management may be called upon to implement ecological restoration or other actions that initiate or inhibit phase shifts. This emphasizes the need for spatially explicit consideration of heterogeneous landscapes, as well as for a robust conceptual model to clearly communicate this information.

STMs allow scientists and managers to synthesize scientific information and clearly communicate among stakeholders what is known about a system, its states, its phases, its thresholds, and its hypothesized drivers of change. They can incorporate spatial heterogeneity and both natural and anthropogenic disturbances. STMs also help identify where knowledge is weak or lacking. Although STMs have been used frequently in rangeland sciences, their application for management in other ecological systems has been limited [10,28]. While they have at times been criticized for not being quantitative (although there is potential (see [29,30]), we argue that given their ability to structure and communicate information about system states, disturbances, and management responses, and to generate hypotheses about drivers and change, STMs are well suited to the conceptual representation of systems undergoing long-term management.

### 1.2.2. Establishing Measurable Objectives

Well-defined objectives and questions are integral to guiding the management and monitoring of ecological systems [8,10,16,24]. Additionally, while STMs help consolidate knowledge into a conceptual representation of a system, they do not necessarily articulate management objectives and questions that can drive a sampling design for monitoring to determine if biodiversity conservation management activities are achieving their desired objectives, and nor do STMs tend to make objectives quantifiable.

Typically, conservation objectives are statements of intent that are then further developed into clear, and ideally measurable, outcomes [25]. Often, there may be questions regarding how to obtain outcomes using management or restoration (e.g., the best methods for nonnative plant species' removal, the effectiveness of planting or thinning treatments to achieve target plant densities, etc.), which require the establishment of monitoring and assessment regimes. Objectives and questions should be germane to management and should help inform the monitoring of specific management actions. They should be collaboratively developed among stakeholders, including scientists, statisticians, natural resource managers, etc. Working to establish clear objectives is especially important when there are multiple, often competing goals for a conservation landscape, for example, allowing for recreational use while protecting and restoring wildlife interactions. In these cases, tradeoffs must often be assessed, and their consequences monitored [25].

However, limited analytical tools are available to assist in (1) identifying target outcomes and questions, (2) identifying specific (preferably quantifiable) metrics to monitor in order to assess progress toward outcomes, and (3) allowing management actions to be

modified in response to information acquired during monitoring in order to improve the probability of attaining outcomes [8]. For the management of rangelands, Derner et al. [8] propose a systematic approach to identifying measurable objectives. Their approach is adaptive and outcome-focused, and emphasizes the need for establishing specific desired outcomes that are quantifiable whenever possible, as well as science-driven monitoring to inform decision making. This facilitates sound data collection that is relevant to management goals. Although focused on grazing management, with modifications, their approach has potential applications in landscape conservation planning and management.

### 1.2.3. Decision Support Tools

While STMs can incorporate landscape heterogeneity, because environmental variation can so strongly influence management actions' success and cost-effectiveness, decision support tools (DSTs) help clarify where, when, and how management actions should be carried out on the landscape [23,31,32]. Over the last two decades, DSTs have gained support in both the government and private sector; however, some have questioned their effectiveness [23,33]. Two criticisms include that they are often used without clearly defined goals and that they need to be adaptive [23,33,34]. DSTs, when designed with clearly articulated goals, can assist land managers in assessing available options under given scenarios and anticipate the potential cost and benefits associated with specific actions [34]. They further explain drivers of system change (between states and phases). They can be used to operationalize STMs and management objectives, articulating the expected outcomes of management actions, and to help identify where experiments are necessary to gather needed information. DSTs should include triggers or thresholds that initiate management actions [22]. When possible, these triggers and thresholds should be quantifiable. Thus, DSTs can be used to directly guide management actions, can ensure management continuity when personnel changes occur, and importantly, they can be adapted when predicted outcomes are shown to be inaccurate, or when new technologies, information, or strategies are obtained. When used together under one systematic structured decision framework, STMs, explicit goal setting with measurable objectives, and decision support tools are complementary and have the potential to facilitate not only the long-term conservation monitoring and management of heterogeneous landscapes, but also, ultimately, a better understanding of how these systems function.

We propose a new framework, L-TEAM (long-term ecological adaptive monitoring), a systematic approach to the long-term adaptive monitoring, management, and evaluation of ecological conservation efforts at the landscape level (Figure 2). Our framework integrates the tools described above that can operationalize Likens and Lindenmayer's (2018) long-term adaptive monitoring approach based on their findings of what makes for successful monitoring projects. L-TEAM integrates STMs, objective oriented goals, and decision support tools as elements that represent system dynamics, define management goals, and guide management actions. The framework helps develop questions and experiments that can test whether goals are met, adapt in light of new information, and support management decisions.

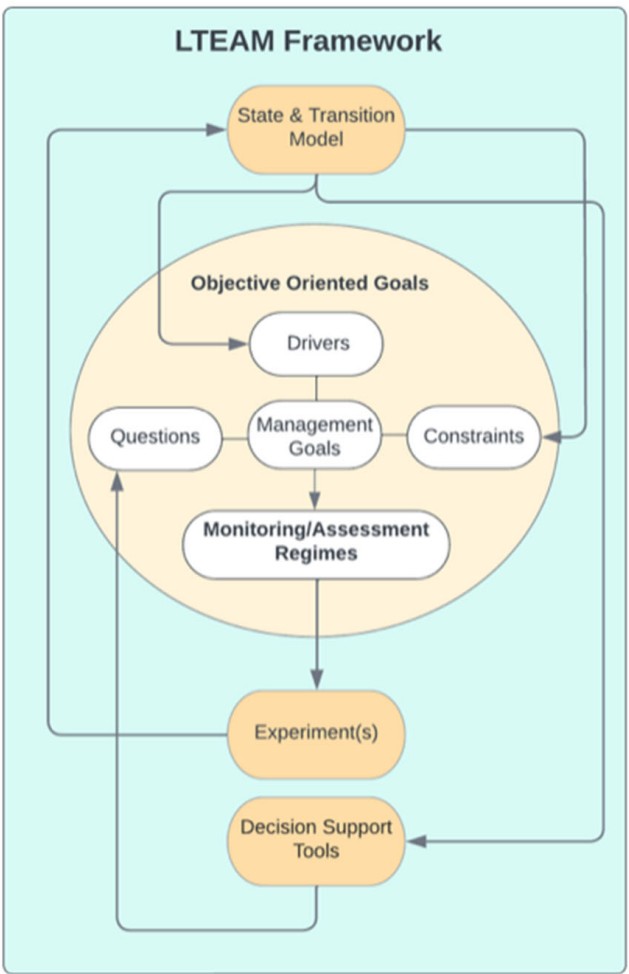

**Figure 2.** A conceptual representation of the Long-Term Ecological Adaptive Monitoring (L-TEAM) framework that depicts how inputs (dark orange ovals) can be used to inform objective-oriented goals (OOGs). The conceptual model is a state-and-transition model, which lends itself well to the representation of complex landscapes. Management and restoration goals are coupled with questions to create objective-oriented goals (OOGs). OOGs articulate the drivers and constraints identified in the STM that directly affect the management goal of concern; they help turn those management goals into questions, and finally they assist in identifying what we need to monitor to answer our questions. This informs the design of rigorous experiments, the results of which can be incorporated back into the STM and then used to inform the development of decision support tools.

## 2. Materials and Methods

To illustrate the use of the L-TEAM framework, we describe its implementation for The Cajon Creek Conservation Area located in California, USA. Cajon Creek represents a conservation-bank approach to biodiversity protection. A conservation bank consists of a parcel or parcels of private property managed in perpetuity for the protection of endangered species. The owner/s of the property/ies are granted credits through state and/or federal agencies for the value of the species and habitat being protected, which they can then use, bank for the future, or sell to offset development [35].

The Cajon Creek Conservation Area (CCCA) is in Riverside County in Southern California (Figure 3), a semiarid region with a Mediterranean-type climate [36]. Cajon Creek vegetation consists primarily of Riversidian Alluvial Fan Sage Scrub (RAFSS), which is a rare Southern Californian alluvial floodplain ecosystem. CCCA hosts about 45 species of conservation concern, including the small mammal *Dipodomys merriami parvus* (San Bernardino Kangaroo Rat; SBKR) and *Eriastrum densifolium* ssp. *sanctorum* (Santa Ana River

Woolly Star), a perennial herb. The RAFFS ecosystem occurs in a highly dynamic alluvial plain of rivers with seasonal flow, and these ecosystems are threatened by development, illegal dumping, invasive species, hydrological modification, and other anthropogenic modifications [36,37].

### Cajon Creek Conservation Area

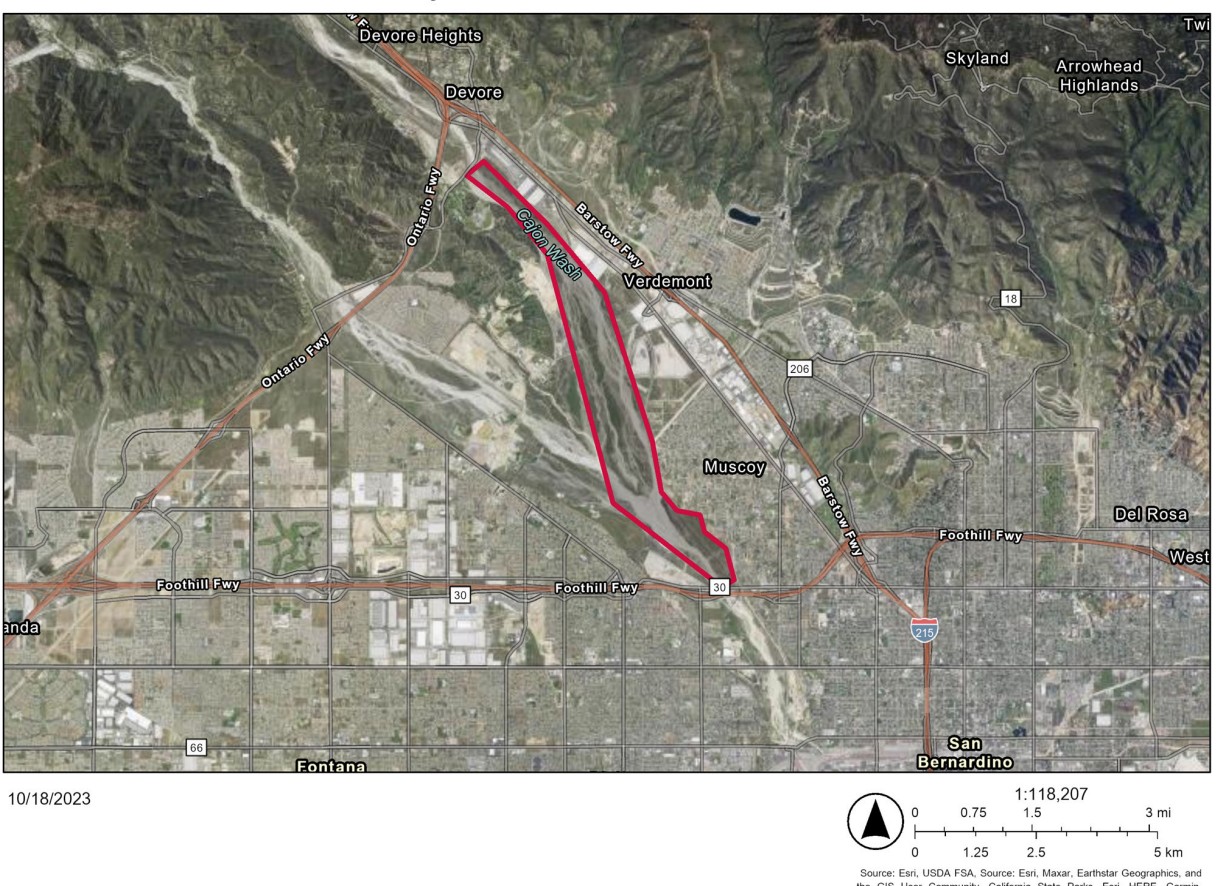

10/18/2023

**Figure 3.** The Cajon Creek Conservation Area (designated by the red border) located in Riverside County in Southern California. The area is a semiarid region with a Mediterranean-type climate [25]. The vegetation consists primarily of Riversidian Alluvial Fan Sage Scrub (RAFSS), which is a rare Southern Californian alluvial floodplain ecosystem.

In Cajon Creek, sand and gravel mining by the Vulcan Materials Company have also had a substantial impact on the system. In 1998, an agreement was established between Vulcan Materials and federal and state agencies to establish the conservation bank and restore and conserve portions of the Cajon Creek property. The conservation area now consists of over 485 ha of pioneer, intermediate, and mature successional phases of RAFSS, mule fat scrub and buckwheat scrub plant communities [36,37]. Conservation efforts rely on ecological restoration of degraded habitat, which to date uses various approaches (grazing, herbicides) to removing nonnative plants, revegetating by imprinting seeding of native plants, and translocating SBKR. Monitoring habitat conditions through established ground-based transects and photo points has been implemented, and annual unmanned aerial vehicle (UAV) flights were recently initiated to collect imagery. A trapping grid is used to monitor SBKR populations.

To develop the STM portion of the L-TEAM framework, we conferred with CCCA stakeholders, regulatory requirements, several years of site documentation, and the published literature on RAFFS systems and SBKR habitat requirements. The system's current states and known and hypothesized drivers and transitions were identified and described. The initial STM can be updated if new information is obtained.

We then met with stakeholders to develop objective-oriented goals (OOGs; the light orange center circle Figure 3). OOGs coupled current management goals with outstanding questions that needed to be answered in order to determine whether goals are being met and what management actions are most effective in meeting those goals. We used the STM to help identify the specific drivers and constraints to achieving specific management goals. The specific drivers and constraints impacting the identified goal were incorporated into each of the OOGs. Finally, the OOGs were used to identify the appropriate monitoring and assessment protocol(s) to answer the outstanding questions and to determine if the goal was achieved. This led us to the development of scientifically sound empirical studies that were designed to answer specific questions and determine if goals have been met.

Working with CCCA stakeholders and reviewing site documentation and the existing literature, we used the results of prior experiments to inform the development of decision support tools. The CCCA STM was used to clearly articulate our hypotheses of which management actions are effective and when and where on the landscape they will be effective. In addition to the STM, the OOGs were used to help identify when and where on the landscape monitoring actions are to take place. Working with stakeholders, we identified thresholds and triggers (either known or hypothesized) that will initiate specific management and monitoring actions.

Notably, the framework is adaptive in that as new information accumulates either through our experiments, consistent monitoring, or from outside sources, we can incorporate it into a revised STM, use it to modify or identify new OOGs, and to update the decision support tools.

## 3. Results

### 3.1. CCCA State-and-Transition Model

The STM for the Cajon Creek Conservation Area highlights the main drivers of floodplain ecosystem dynamics: flood events and time since the last flood event (Figure 4). Floods scour existing vegetation and soil, transitioning parts of the landscape back to an earlier phase [36,37], resulting in a mosaic of habitats. Elevation and soil are also key factors; typically, higher elevations have more developed soils and vegetation phases. Within the successionally intermediate RAFSS phase, we also identified three shrub cover levels related to conservation management goals (lower cover has higher habitat quality for SBKR). We also identified two (undesirable) alternative vegetation states: nonnative annual grasslands cover, consisting primarily of *Avena* and *Bromus* spp.; and nonnative herbaceous cover, consisting primarily of *Brassica* spp. We identified state transitions and their drivers, and transient transitions between states and phases, as well as the hypothesized management actions (e.g., conservation grazing, manual removal) necessary to reverse, initiate, or prevent undesirable transitions (Table 2).

### 3.2. OOGS

For CCCA, an overarching goal is the restoration of habitat for species of conservation concern. Studies have shown that both SBKR and Woolly Star prefer a more open shrub habitat, with some studies suggesting over 60% shrub cover is too dense, and cover of 30% or less as ideal [38]. However, there is some concern that reducing cover too much may invite invasion by nonnative plant species. The drivers related to this habitat conservation goal (Figure 5A) identified in the STM (Figure 4) are time since flood event and soil texture. Some of the constraints to both monitoring and attaining this goal are the size of the site (large sites may be too costly to monitor through ground-based transects), the limited ability to manipulate hydrology and soil texture, and potential invasion by nonnative species. This management constraint initiated the question of whether cover of intermediate RAFFS could be determined by monitoring through imagery obtained by unmanned aerial vehicles. Managers are also interested in determining how much cover should be reduced (to 20–30% or 30–40%) in order to promote occupancy by the species of concern, while not inviting encroachment by nonnatives (Figure 5A). To address these

questions and assess progress toward management goals, working with stakeholders, we determined appropriate monitoring protocols that included the initiation of strategic UAV flights with appropriate imagery acquisition and analysis, continued SBKR trapping, and assessments of both the effectiveness of shrub cover reduction activities and the effects of shrub cover reduction on SBKR occupancy and the presence of nonnatives. This led to the development of the experiments discussed below.

**Table 2.** CCCA transitions and hypothesized management actions.

| T (Transitions) | TT (Transient Transitions) | HM (Management Actions Hypothesized to Initiate Transitions) |
|---|---|---|
| **T1:** From Pioneer RAFFS to nonnative grasslands. Clearing/mining. Roads. Unauthorized access (humans and horses). Fire. Introductions of invasive species. | **TT1:** from pioneer RAFFS to intermediate RAFFS. Time since flood event and elevation. | **HM1:** actions to convert nonnative grasslands cover to pioneer RAFFS: restrict access, herbicide, targeted grazing. |
| **T2:** From intermediate RAFSS to nonnative herbs/forbs. Clearing/mining. Roads. Introduction of nonnative species. Precipitation | **TT:** from intermediate RAFFS to mature RAFFS. Time since flood event and elevation. | **HM2:** Actions to remove nonnatives from intermediate RAFFS: restrict access, spot herbicide., targeted grazing. |
| | | **HM3:** Actions to maintain vegetation cover of intermediate RAFFS < 30%: manual removal, spot herbicide, hydrological manipulation. |
| | | **HM4:** Actions to maintain plant cover of intermediate RAFFS < 30%: manual removal, spot herbicide, hydrological manipulation. |
| | | **HM5:** Actions to transition intermediate RAFFS to Pioneer: scour via mechanical removal or hydrologic manipulation. |

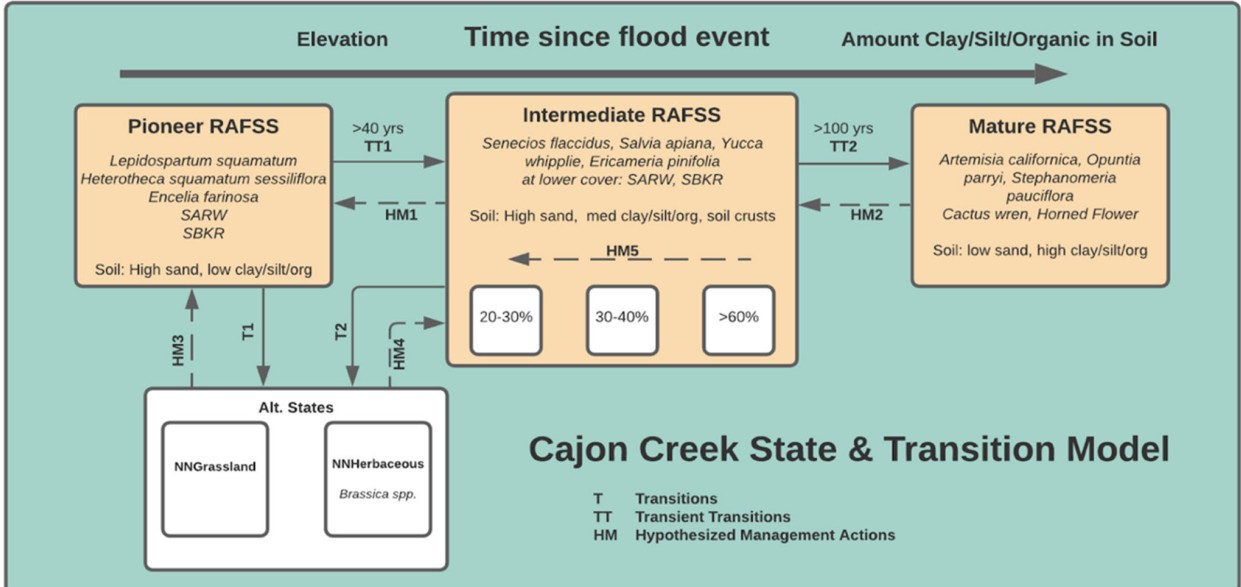

**Figure 4.** This is the schematic portion of the state-and-transition model developed for the Cajon Creek Conservation Area. Some of the main drivers of the system are flood events and time since the last event. Elevation and soil are also key factors. Within the intermediate RAFSS (Riversidian Alluvial Fan Sage Scrub) habitat type, three cover levels related to management goals were identified. Two alternative vegetation states were also identified:, nonnative grasslands and nonnative herbaceous cover. Transitions (T) and transient transitions (TT) were identified, as well as the hypothesized management actions (HM) necessary to reverse, initiate, or prevent transitions (see also Table 2). SAWR: Santa Ana River Woolly Star; SBKR: San Bernardino Kangaroo Rat (the two focal conservation target species).

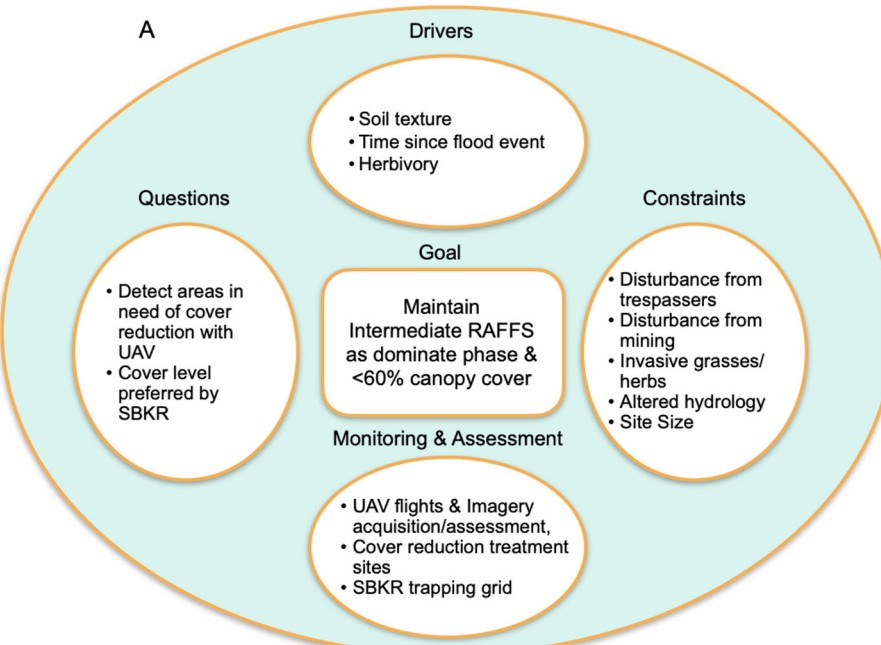

Treatment Plots Within Each Experimental/Treatment Site

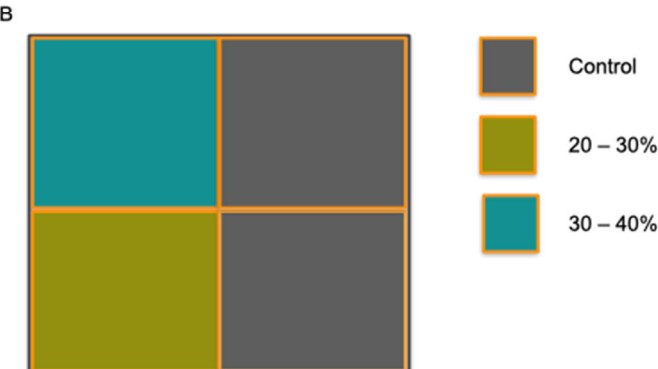

**Figure 5.** One of the objective-oriented goals (**A**) and a schematic of the thinning treatment sites (**B**) developed with stakeholders for the Cajon Creek Conservation Area. Goal: Studies indicate that SBKR prefers intermediate RAFFS vegetation with more open canopies. Monitoring/Assessment: To maintain vegetation in this phase, the site must be monitored and thinning actions performed when necessary. Use of UAVs has the potential to support comprehensive, efficient monitoring if it is possible to detect RAFFS phases and cover amounts. Questions: Additionally, it will be important to determine which thinning level is most appropriate, resulting in increases in SBKR habitat use while still inhibiting invasion by nonnative grasses/forbs. Experimental thinning sites were established. At each site, subplots consisted of controls and subplots manually thinned to 20–30% and 30–40% cover.

### 3.3. Experiments

Based on the CCCA OOG (Figure 5), experiments were designed to assess whether UAV monitoring can be used to identify the different cover ranges in the intermediate RAFFS cover phase of the RAFFS state, and whether cover reduction by manual removal promotes SBKR occupancy. UAV flights encompassing restoration sites were established and are to be flown annually. The UAV flight path includes treatment sites that were established in intermediate RAFSS habitats consisting of two levels of cover reduction via manual removal: reduction to 20–30% and 30–40%–and control plots (Figure 5B). The UAV imagery includes visible (red–green–blue) and near-infrared (NIR) wavelength bands in order to calculate the normalized difference vegetation index (NDVI) from red and NIR

reflectance, and enhance the detection of green vegetation [39]. Imagery is being assessed to determine which spectral information is best able to discriminate the different cover amounts (see example Figure 6). Furthermore, taking advantage of a previously established trapping grid for SBKR, treatment sites were located to incorporate portions of the trapping grid; thus, SBKR occupancy can be compared between treatment plots, and used to control (untreated) areas located outside of the treatment sites. These experiments are ongoing; thus, we do not present their results here.

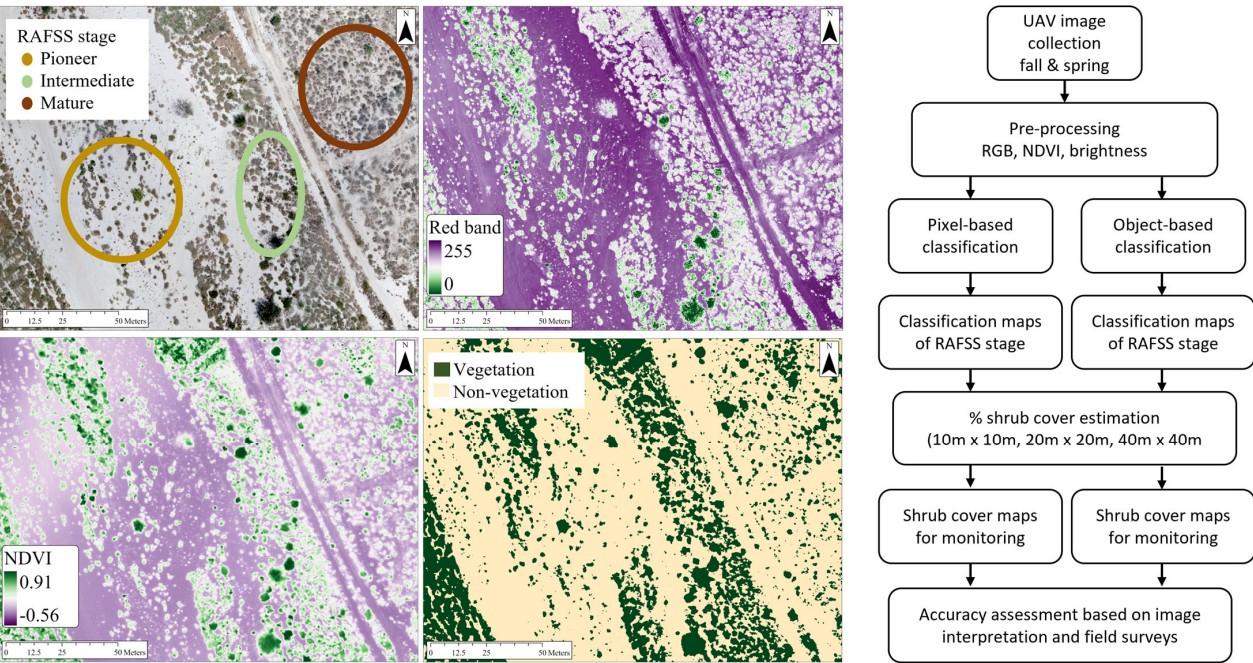

**Figure 6.** Example of image analysis of Cajon Creek Conservation Area shrub cover monitoring, showing initial imagery from the September 2021 flight. Methods are based on those of Warkentin et al. [40]. Panchromatic imagery showing part of the study area (**upper left**); red bands (**upper right**) and NDVI (**lower left**) for the same area; classification of vegetation/non-vegetation to be used to estimate shrub cover (**lower right**). The workflow (**left panel**) describes the various steps from the UAV imagery to shrub cover assessments. Figure developed by M.B. Rose.

### 3.4. Decision Support Tools

We used the STM (Figure 4) and OOGs (Figure 5) to develop the decision support tool for CCCA (Figure 7). Thresholds were identified that trigger management actions (e.g., nonnative plant cover of greater than 10%). The three parts of this decision support tool (Figure 7A–C) correspond to goals, drivers and constraints defined in the OOG: (A) the RAFFS phase for the habitat under consideration; (B) the annual precipitation (as precipitation and flood events are system drivers); and (C) the functional group of nonnative herbaceous plants present (broadleaf forbs versus graminoids). For example, if the intermediate RAFFS has more than 60% cover, spot treatment or mechanical removal of dense native vegetation is to be initiated. During a wet year, UAV imagery should be used to assess damage to the channel, and spring flushes of nonnative herbaceous vegetation cover and removal of nonnative vegetation should be initiated before seed-setting, via herbicide for grasses and sheep grazing for forbs when still green. As new information is obtained, the decision support tool can be updated, thus allowing for data collected through long-term monitoring to directly influence management actions.

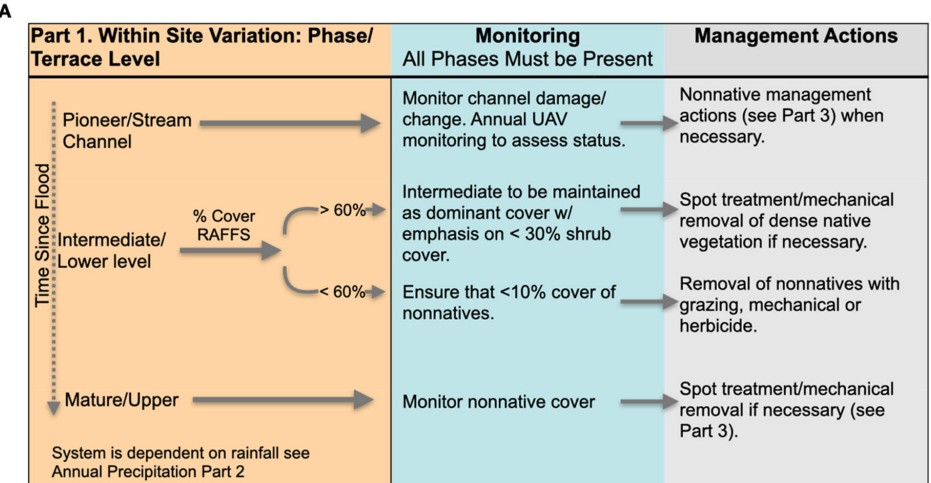

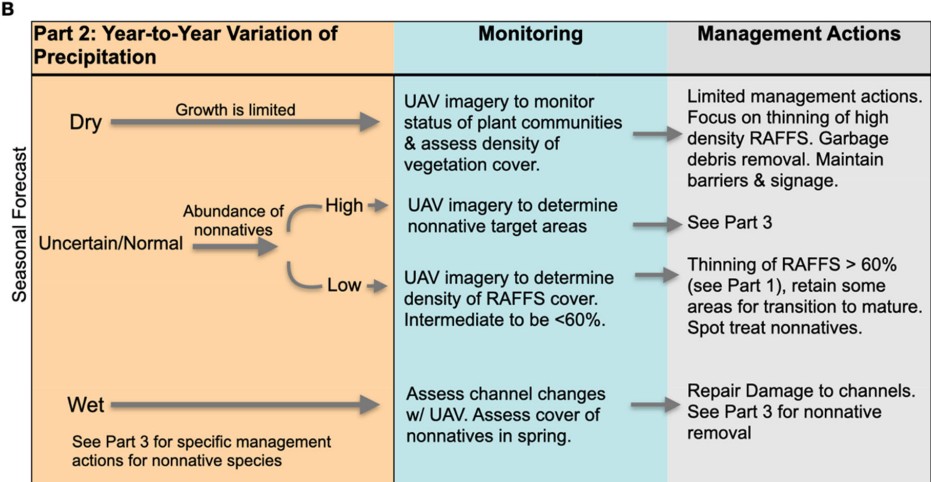

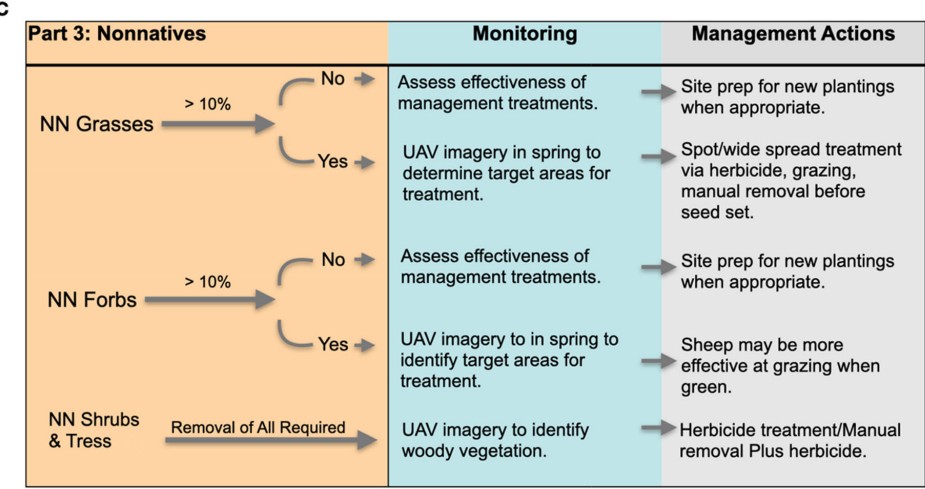

**Figure 7.** A summary of the decision support tool developed with stakeholders for the Cajon Creek Conservation Area, simplified for publication purposes (adapted from Kimball et al. [22]). Management goals and the system's STM informed the design of the decision support tool to help guide management actions and ensure management continuity. Thresholds are included that trigger management actions. The tool describes the different management actions necessary for (**A**) the particular phase/habitat under consideration, (**B**) responding to variation in annual precipitation (as precipitation and flood events are system drivers), and (**C**) the functional group of nonnatives present.

## 4. Discussion

Long-term ecological management and monitoring in support of conservation at the landscape level is challenging. Likens and Lindenmayer [10] and others [11–14] have noted that programs that fail often lack well-defined goals and questions, robust conceptual models, sound experimental designs, and strategic plans to adapt to change and new information. The L-TEAM framework is a systematic approach to addressing these challenges by (1) clarifying understanding of ecosystem function and drivers through the use of STMs; and (2) assisting stakeholders through the use of objective-oriented goals in establishing actionable goals that can be assessed through the development of rigorous experiments, the results of which inform further understanding of system function and drivers. These goals can then be used to (3) develop decision support tools to support management decisions and actions; (4) L-TEAM is adaptive in that it provides a strategic, iterative approach with allows for adaptation to change and the ability to update when new information is obtained. Below, we address some of the challenges that remain, and where future applications and improvements of the L-TEAM framework should consider ways to mitigate these challenges.

It is essential that communication among stakeholders is maintained, especially among conservation land managers and researchers. Input from landowners/managers must inform both the development and implementation of the framework [41,42]. The process should be iterative to ensure that goals are clear and agreed upon, implementation of management actions and experiments are accurate, and results are understood. While L-TEAM is designed to facilitate clear, open, and iterative communication, it remains the responsibility of the stakeholders to put this communication into practice. This requires regular interactions, especially during development. However, as time passes, it is also important that communication channels remain open, especially when there is turnover in individuals in an organization. Easy, live access to L-TEAM documentation and regular reviews (i.e., a living document in the cloud) may further ensure clear communication.

While L-TEAM is adaptive, uncertainty still looms large. Project goals may change radically in the face of unexpected disturbance, climate change, or sale of the land and/or management change. Because the L-TEAM framework is based on an adaptive, iterative processes, uncertainties and broader-scale events, such as global climate change, have the potential to be incorporated into the system as the long-term monitoring that L-TEAM supports provides data on how these changes may be affecting a landscape. Although L-TEAM is designed to be updated and adapted to reflect these changes, it has yet to be tested under such circumstances. However, these uncertainties also bring with them opportunities for further understanding of how to manage and monitor for conservation in heterogeneous landscapes under uncertainty. In fact, long-term monitoring is one of the best tools we have to capture these potential changes and shifts in baselines [10,13]. Thus, L-TEAM, with its emphasis on STMs and OOG development, is not only uniquely positioned to detect the effects of broader scale drivers; it is also well suited—through its adaptive approach and emphasis on identifying questions and applying a rigorous experimental design to answer questions—to supporting management responses in a systematic and scientifically informed manner.

The development of new technologies for monitoring and management is both exciting and challenging. It is important when considering the adoption of new technologies to be cognizant of how this will affect the continuity of data collection and analyses moving forward. While it may be tempting to implement the latest innovations (e.g., UAV-borne imagery, field-based environmental sensors, machine learning for data mining), maintaining the integrity of the monitoring and assessment program should take precedence. Furthermore, new technologies can be expensive, and require new infrastructure and training. The L-TEAM framework's emphasis on objective-oriented goals and the identification of key questions and designs of experiments to answer questions and assess goal attainment should help in determining when and if new technologies should be adopted; in fact, it can help determine if they are beneficial.

Related to issues with technological advancements are those concerning data management, storage, and curation [10,43]. Long-term management and monitoring can result in copious amounts of data. While L-TEAM advocates for data collection, it does not explicitly address how these data should be managed. The ability to organize, store, and readily access large amounts of data is becoming more accessible and affordable; however, it can still prove prohibitive for some projects and organizations. Moreover, special training may be required, and not all organizations have the team members or resources for such training. Best practices are available [44,45], and, when possible, at least one project member should be familiar with these practices. This problem is not unique to landscape conservation management planning. However, L-TEAM's emphasis on clearly identifying specific factors to monitor should help, to some extent, by tempering the tendency to monitor a "blizzard of details" [10], thus generating an abundance of irrelevant data. Additionally, L-TEAM's documentation of experiments should ensure that protocols for data collection maintain integrity, unless stakeholders agree upon changes. Moving forward, the L-TEAM framework may be improved by including consideration of data management and a way to clearly articulate standards for data collection, curation, and best practices.

Despite these challenges, L-TEAM offers a toolbox for a more systematic approach to long-term ecological management and monitoring at the landscape level. L-TEAM combines STMs, objective-oriented goals, and decision support tools into a framework that can help scientists and managers design a long-term monitoring and management plan for landscape projects. Future evaluation of L-TEAM should include its application in different contexts. L-TEAM also lends itself well to incorporating diverse stakeholder knowledge and management strategies; for example, it has the potential to include indigenous knowledge and management practices [46,47]. It is especially well suited to capturing the complexity inherent in landscape-level projects, which includes habitat heterogeneity, multiple land uses and land use histories, and multiple management goals, including conservation. L-TEAM guides the establishment of sound experiments to answer well-defined questions that will help not only in improving the management of complex systems, but in our overall understanding of how these systems function and their responses to management actions through an adaptive approach.

**Author Contributions:** Conceptualization, M.M., J.F. and L.L.; methodology, M.M.; writing—original draft preparation, M.M.; writing—review and editing, M.M., J.F. and L.L.; visualization, M.M.; supervision, J.F. and L.L.; All authors have read and agreed to the published version of the manuscript.

**Funding:** This research was supported in part by a research award from the Vulcan Materials Company Foundation (to L.L.), and by a University of California Eugene Cota Robles Scholarship, (to M.M.). The APC was funded by San Diego State University (J.F.).

**Data Availability Statement:** No new data were created or analyzed in this study. Data sharing is not applicable to this article.

**Acknowledgments:** We thank M. Brooke Rose for developing Figure 7. We thank Vulcan Materials company for their support, funding, and access to the land and documents. We especially thank Tom McGill and Sharon Lockhart for answering our many questions and participating in the development of CCCA's L-TEAM.

**Conflicts of Interest:** The authors declare no conflict of interest. The funders had no role in the design of the study; in the collection, analyses, or interpretation of data; in the writing of the manuscript; or in the decision to publish the results.

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
