# Peer review of "Enhancing the Long-Term Ecological Management and Monitoring of Landscapes: The L-TEAM Framework"

_land, doi:10.3390/land12101942_

Round 1
Reviewer 1 Report
In this paper, the authors address the hurdles faced in sustaining long-term monitoring and management for landscape-scale efforts by offering three promising conceptual and methodological developments that support such initiatives. Then they introduce L-TEAM, a Long-Term Ecological Adaptive Monitoring and Management framework, that integrates those three components using four tools -- a robust conceptual model, clearly defined measurable goals, rigorous experimentation, and decision support tools. Finally, using a case study, L-TEAM's effectiveness in supporting long-term monitoring and management of a landscape conservation project with heterogeneous habitats and varying management objectives is demonstrated.
The identified criteria for the proper management and spread of landscapes are clear, obvious and not very revealing. It is possible that the current state of research is not something new. The authors provided little definition of what is new in their work, and did not emphasize the strengths and weaknesses of their concept.
It is worth mentioning that the adoption of specific paradigms (functionalism/structuralism) is important in landscape monitoring and management, as is the verification of the correctness of monitoring and management procedures, allowing for the detection of errors in assessment, carelessness, the use of inappropriate tools, etc.
Depending on the assumptions made (what are we protecting? Species or process?), the landscape management and monitoring procedure will differ.
The authors emphasize the important role of the tools used in monitoring (subsection 1.3.3), but there is no discussion of the tools themselves. Rather, they focus on action patterns, procedures and goals.
The authors present their concept based on a case study - coastal scrub in California. However, there is not much information here about what exactly was implemented there in practice, whether monitoring has been ongoing, for how many years, and has it proven to be effective? Or have the authors just developed an action program that will be implemented?
The work requires serious editorial correction. There is a lot of chaos there both in the numbering of chapters, the presentation of content and figures, the list of bibliographies, etc.
Detailed notes:
The text is carelessly prepared and requires careful editing.
There are double spaces and double dashes in the abstract
The term "keyword 1" appears in keywords (line 25)
In the introduction (chapter Introduction), the same text is repeated in lines 29-34 and 35-40.
In chapter 1 - no subchapter 1.1
In chapter 3, before subchapter 3.1, tables and figures appear without any introduction. Why are they placed in a chapter instead of in subchapters?
Figure 6- is illegible
The bibliography is written differently (year in brackets after the name - in literature items no. 1, 8, 10, 20)
Notes to review of the literature:
(lines 126-134) – it is worth emphasizing that environmental disturbances do not always lead to landscape degradation. They are often the basis for succession and better adaptation of the phytocoenosis, and even determine the achievement of climax.
In chapter 1.3.3. no discussion of the tools used in monitoring. There are only generally discussed methods, methods and procedures of monitoring. Meanwhile, this thread appears only in the discussion chapter. The authors discuss new technologies that have not been discussed before either in the literature review or in their own research.
In the materials and methods chapter - there is no detailed information on the principles of implementation of the concept proposed by the authors in the case study - coastal scrub in California
The work requires major changes and a second review to be published.
Author Response
Reviewer #1 Comments and Suggestions for Authors
In this paper, the authors address the hurdles faced in sustaining long-term monitoring and management for landscape-scale efforts by offering three promising conceptual and methodological developments that support such initiatives. Then they introduce L-TEAM, a Long-Term Ecological Adaptive Monitoring and Management framework, that integrates those three components using four tools -- a robust conceptual model, clearly defined measurable goals, rigorous experimentation, and decision support tools. Finally, using a case study, L-TEAM's effectiveness in supporting long-term monitoring and management of a landscape conservation project with heterogeneous habitats and varying management objectives is demonstrated.
The identified criteria for the proper management and spread of landscapes are clear, obvious and not very revealing. It is possible that the current state of research is not something new. The authors provided little definition of what is new in their work, and did not emphasize the strengths and weaknesses of their concept.
R1 RESPONSE 1: We have emphasized what is new in our work in the following text, for example:
Introduction: “...there has yet to be clear guidance on how to implement an adaptive monitoring program using a standardized approach – especially at the landscape level… Here we present a systematic, holistic approach to developing long-term ecological adaptive monitoring and management (L-TEAM).”
Discussion: “Long-term ecological management and monitoring in support of conservation at the landscape-level is challenging, and L-TEAM helps address challenges related to defining goals, formulating questions, designing monitoring experiments and adapting to change.” “...L-TEAM is offered as a toolbox for a more systematic approach to long-term ecological management and monitoring at the landscape-level.”
Further we have added the following text (Lines 126-129): “Then, we integrate these procedural and analytical tools for the first time into a novel framework for applying principles from long-term ecological monitoring and management to the practice of managing heterogeneous landscapes to meet conservation goals. “
We note however that this overall assessment that the framework we propose does not offer something new is at odds with the assessment of the Editor and Reviewer 2.
Reviewer #1 It is worth mentioning that the adoption of specific paradigms (functionalism/structuralism) is important in landscape monitoring and management, as is the verification of the correctness of monitoring and management procedures, allowing for the detection of errors in assessment, carelessness, the use of inappropriate tools, etc. Depending on the assumptions made (what are we protecting? Species or process?), the landscape management and monitoring procedure will differ.
R1 RESPONSE 2: Exactly. The management and monitoring procedures would differ, but the framework for determining what those procedures are can be applied in either case. Whether landscape management goals are defined in terms of structural or functional attributes, for example, species diversity and vegetation cover, or hydrological regime and primary productivity, the L-TEAM framework can be used to identify the procedures that are appropriate to the goals. Not only that, the L-TEAM framework can help identify whether structural or functional targets are appropriate in light of thoughtfully defined management objectives. This paper does not address specific monitoring targets or data collection procedures – rather it provides a structured decision framework for identifying system components and management goals, from which monitoring targets and management procedures can be derived, and through which the efficacy of procedures can be evaluated (through experiments and adaptive monitoring and management).
Reviewer #1 The authors emphasize the important role of the tools used in monitoring (subsection 1.3.3), but there is no discussion of the tools themselves. Rather, they focus on action patterns, procedures and goals.
R1 RESPONSE 3: We think there may be a semantic misunderstanding or disagreement here. We note that Section 1.2.3 describes the development of decision trees that serve as decision support tools for monitoring and management. The term ‘decision support tool’ is well established and widely used in the literature in many disciplines (for example: Roesch-McNally et al. 2021). Furthermore, we build on this established concept and refer to all four components of our L-TEAM framework as ‘tools’ and the framework as a ‘toolbox.’ For example, STMs are tools used to define the phases, states and transitions in a complex system. We think the term is appropriate in this context, as do the Editor and other reviewers. We disagree that ‘tool’ can only refer to the gadgets, technologies and methodologies used to monitor and manage (UAVs, goats, tape measures, eDNA, trapping grids, questionnaires…) and our meaning of procedural and analytical tools and toolbox was quite clear to other reviewers and editors.
Reviewer #1 The authors present their concept based on a case study - coastal scrub in California. However, there is not much information here about what exactly was implemented there in practice, whether monitoring has been ongoing, for how many years, and has it proven to be effective? Or have the authors just developed an action program that will be implemented?
R1 RESPONSE 4: We provided extensive background information on the case study which takes place in an Alluvial Fan Sage Scrub (not coastal) community. We stated: “In Cajon Creek, sand and gravel mining by the Vulcan Materials Company has also had substantial impacts on the system. In 1998, an agreement was established between Vulcan Materials and federal and state agencies to establish the conservation bank and restore and conserve portions of the Cajon Creek property. The Conservation Area now consists of over 485 ha of pioneer, intermediate, and mature successional phases of RAFSS, mule fat scrub and buckwheat scrub plant communities [25,26]. Conservation efforts rely on ecological restoration of degraded habitat which to date uses various approaches to removing nonnative plants, revegetating by imprinting seeding of native plants, and translocating SBKR. Monitoring habitat conditions through established ground-based transects and photo points has been implemented and annual Unmanned Aerial Vehicle (UAV) flights were recently initiated to collect imagery. A trapping grid is used to monitor SBKR populations.
To develop the STM portion of the L-TEAM framework, we conferred with CCCA stakeholders, regulatory requirements, several years of site documentation, and existing literature on RAFFS systems and SBKR habitat requirements. The system’s current states and known and hypothesized drivers and transitions were identified and described.”
If the reviewer is referring specifically to the framework for monitoring and management presented in the Results section of the paper, that action program was only recently developed by the authors in consultation with the site managers and so it has not yet been implemented over many years. We state this in the paper, for example: “Furthermore, taking advantage of a previously established trapping grid for SBKR, treatment sites were located to incorporate portions of the trapping grid – thus, SBKR occupancy can be compared between treatment plots, and to control (untreated) areas located outside of the treatment sites. These experiments are ongoing thus we do not present their results here.”
Reviewer #1 The work requires serious editorial correction. There is a lot of chaos there both in the numbering of chapters, the presentation of content and figures, the list of bibliographies, etc.
R1 RESPONSE 5: We have carefully checked everything in our revision. We note that some of these errors seem to have been introduced by the journal’s publisher, as a result of difficulties using their manuscript template.
Reviewer #1 Detailed notes: The text is carelessly prepared and requires careful editing. There are double spaces and double dashes in the abstract
R1 RESPONSE 6. All double spaces and dashes have been removed from the abstract.
Reviewer #1 The term "keyword 1" appears in keywords (line 25)
R1 RESPONSE 7. Removed
Reviewer #1 In the introduction (chapter Introduction), the same text is repeated in lines 29-34 and 35-40.
R1 RESPONSE 8. Repeated lines have been deleted.
Reviewer #1 In chapter 1 - no subchapter 1.1
​​R1 RESPONSE 9. Subchapters within chapter one have been adjusted.
Reviewer #1 In chapter 3, before subchapter 3.1, tables and figures appear without any introduction. Why are they placed in a chapter instead of in subchapters?
R1 RESPONSE 10. All figures within chapter 3 have been moved within the appropriate subchapter and are now beneath the paragraph they are mentioned within.
Reviewer #1 Figure 6- is illegible
R1 RESPONSE 11. Figure has been updated with a higher resolution image.
Reviewer #1 The bibliography is written differently (year in brackets after the name - in literature items no. 1, 8, 10, 20)
R1 RESPONSE12. Bibliographic references have been updated to reflect suggestions.
Reviewer #1 Notes to review of the literature: (lines 126-134) – it is worth emphasizing that environmental disturbances do not always lead to landscape degradation. They are often the basis for succession and better adaptation of the phytocoenosis, and even determine the achievement of climax.
air and active restoration is now required…” This refers to degradation of key ecological processes required to maintain system integrity – it does not refer toR1 RESPONSE13: We agree that disturbance is an important ecological process. The text referred to does not say that landscape disturbance leads to land degradation. This paragraph gives well established definitions of the components of a STM and says “Thresholds are points in space and time where, once crossed, the key ecological processes responsible for a system's identity degrade past a point of self-repair natural disturbance or to degradation of the system. In fact, natural disturbance is an example of such a key ecological process. We furthermore state: “Moreover, if natural disturbances (for example, a natural flooding regime) are no longer operating to drive shifts between phases, management may be called upon to implement ecological restoration or other actions that initiate or inhibit phase shifts.” We added: “Unlike state changes, phase-shifts align with a site’s natural disturbance regime and (successional) trajectory – they fall within the site’s “identity”.
And we added Lines 166-168: Transient and irreversible transitions are often triggered by natural or human-caused disturbances which may occur quickly, as with a fire or flood, or more slowly in response to repeated stress such as grazing or drought.
Reviewer #1 In chapter 1.3.3. no discussion of the tools used in monitoring. There are only generally discussed methods, methods and procedures of monitoring. Meanwhile, this thread appears only in the discussion chapter. The authors discuss new technologies that have not been discussed before either in the literature review or in their own research.
R1 RESPONSE 14: We have already responded to this point, above. Please see R1 Response 3.
Reviewer #1 In the materials and methods chapter - there is no detailed information on the principles of implementation of the concept proposed by the authors in the case study - coastal scrub in California
R1 RESPONSE 15: We did provide this information. We stated: “Working with CCCA stakeholders and reviewing site documentation and existing literature, we used the results of prior experiments to inform the development of Decision Support Tools. The CCCA STM was used to clearly articulate our hypotheses of which management actions are effective and when and where on the landscape they will be effective. In addition to the STM, the OOGs were used to help identify when and where on the landscape monitoring actions are to take place. Working with stakeholders, we identified thresholds and triggers (either known or hypothesized) that will initiate specific management and monitoring actions.”

Reviewer 2 Report
See MS WORD attachment

As suggested in the attached review, the language of the Abstract could be made more readable for a wider audience that includes non-scientists.
